# Common and distinct neural representations of aversive somatic and visceral stimulation in healthy individuals

Lukas Van Oudenhove[1,2,3,13], Philip A. Kragel [4,12,13], Patrick Dupont [5], Huynh Giao Ly[1], Els Pazmany[6], Paul Enzlin[6], Amandine Rubio[7], Chantal Delon-Martin [7], Bruno Bonaz[7], Qasim Aziz[8], Jan Tack[9], Shin Fukudo [10], Michiko Kano[10,11] & Tor D. Wager [3✉]

Different pain types may be encoded in different brain circuits. Here, we examine similarities and differences in brain processing of visceral and somatic pain. We analyze data from seven fMRI studies ($N = 165$) and five types of pain and discomfort (esophageal, gastric, and rectal distension, cutaneous thermal stimulation, and vulvar pressure) to establish and validate generalizable pain representations. We first evaluate an established multivariate brain measure, the Neurologic Pain Signature (NPS), as a common nociceptive pain system across pain types. Then, we develop a multivariate classifier to distinguish visceral from somatic pain. The NPS responds robustly in 98% of participants across pain types, correlates with perceived intensity of visceral pain and discomfort, and shows specificity to pain when compared with cognitive and affective conditions from twelve additional studies ($N = 180$). Pre-defined signatures for non-pain negative affect do not respond to visceral pain. The visceral versus the somatic classifier reliably distinguishes somatic (thermal) from visceral (rectal) stimulation in both cross-validation and independent cohorts. Other pain types reflect mixtures of somatic and visceral patterns. These results validate the NPS as measuring a common core nociceptive pain system across pain types, and provide a new classifier for visceral versus somatic pain.

[1] Laboratory for Brain-Gut Axis Studies (LaBGAS), Translational Research Center for Gastrointestinal Disorders (TARGID), Department of Chronic Diseases, Metabolism, and Ageing, KU Leuven, Leuven, Belgium. [2] Leuven Brain Institute, KU Leuven, Leuven, Belgium. [3] Cognitive and Affective Neuroscience Lab, Department of Psychological and Brain Sciences, Dartmouth College, Hanover, NH, USA. [4] Department of Psychology and Neuroscience and the Institute of Cognitive Science, University of Colorado Boulder, Boulder, CO, USA. [5] Laboratory for Cognitive Neurology, Department of Neurosciences, KU Leuven, Leuven, Belgium. [6] Interfaculty Institute for Family and Sexuality Studies, Department of Neurosciences, University of Leuven, Leuven, Belgium. [7] Grenoble Institute of Neuroscience, University of Grenoble Alpes, Grenoble, France. [8] Barts and The London School of Medicine and Dentistry, Wingate Institute of Neurogastroenterology, Centre for Neuroscience and Trauma, Blizzard Institute, Queen Mary University of London, London, UK. [9] Gastrointestinal Motility and Sensitivity Research Group, Translational Research Center for Gastrointestinal Disorders (TARGID), Department of Chronic Diseases, Metabolism, and Ageing, KU Leuven, Leuven, Belgium. [10] Behavioral Medicine, Graduate School of Medicine, Tohoku University, Sendai, Japan. [11] Frontier Research Institute for Interdisciplinary Sciences (FRIS), Tohoku University, Sendai, Japan. [12] Present address: Department of Psychology, Emory University, Atlanta, GA, USA. [13] These authors contributed equally: Lukas Van Oudenhove, Philip A. Kragel. ✉email: tor.d.wager@dartmouth.edu

Pain is a primary force motivating human behavior, learning, and neuroplasticity. It is defined primarily as an aversive experience[1] that arises from its signal value as an indicator of current or future bodily harm[2]. But a century of research on conditioning and reinforcement learning demonstrates that pain is more than a momentary experience. The danger signals underlying pain drive neuroplastic changes in the central nervous system that mediate escape, avoidance, and other defensive behaviors[3,4]. However, despite important progress, our mechanistic understanding of pain and its brain bases remain incomplete[5].

One crucial gap relates to the brain representations underlying different types of pain. Neurological and functional differences may track differences in etiology, qualities, or the types of tissues affected, which have different implications for the organization of motivated behavior. However, most mechanistic pain studies in humans, including functional neuroimaging studies, have been limited to cutaneous somatic stimulation, mainly on the hand or foot, as a way of evoking pain[6]. Visceral pain—pain arising from soft tissues and internal organs—is particularly understudied, despite it being one of the most common disease-related forms of pain (e.g., in irritable bowel syndrome (IBS)) and one of the primary causes for seeking medical attention[7].

Several important differences between somatic (cutaneous or musculoskeletal) and visceral pain have led researchers to believe that they constitute fundamentally different types of pain. First, visceral pain has different sensory characteristics: it is less clearly localized, less well correlated with objective stimulation intensity, and very often referred to other body parts[8]. Second, visceral pain is thought to have a stronger affective dimension (reflected in higher unpleasantness relative to intensity ratings), though evidence supporting this hypothesis is limited[9,10]. Differences in afferent input[8] and/or central brain representations may account for these differences in sensory and affective perceptual features. One common assumption is that somatic and visceral pain are processed in different systems—the "lateral pain system" (lateral thalamus, somatosensory cortex, posterior insula) and "medial pain system" (medial thalamus, anterior cingulate/medial prefrontal cortex), respectively. However, definitive evidence for this distinction has been elusive. Despite the relevance of this knowledge gap, only a few studies in small samples of healthy controls have compared visceral and somatic pain directly using functional brain imaging. These studies have yielded mixed findings, including similarities as well as differential responses in somatosensory, cingulate, insular, and prefrontal cortices[10–14] and in subcortical regions, including hippocampus and dorsal pons[10].

Recently, building on advances in multivariate brain models to predict pain and other perceptual outcomes[15–18], Wager et al. used functional magnetic resonance imaging (fMRI) to identify a spatially distributed pattern of brain responses that predict the perceived intensity of experimentally induced thermal somatic pain in healthy individuals. This model is referred to as the "neurologic pain signature" (NPS)[19] to provide a label for testing in new samples and studies. The NPS includes positive predictive weights (i.e., increased response predicts increased pain) in the thalamus, anterior insula and frontal operculum, posterior insula, and secondary somatosensory cortex, anterior midcingulate cortex (aMCC), and lateral parietal cortex. It also includes negative predictive weights (i.e., increased activity is associated with reduced pain) in the ventromedial prefrontal cortex (vmPFC), lateral temporal cortex, occipital cortex, and medial parietal cortex, including posterior cingulate cortex (PCC) and precuneus. The NPS has now been evaluated in over 40 published study cohorts (for reviews, see refs. [20,21]). Several studies of conceptually similar yet nonpainful affective processes have indicated

that the NPS is specific to somatic pain when compared with social rejection, negative emotion, and vicarious pain ("pain" felt when viewing images of others in pain). Each of these conditions has a distinct, reliable neural measure that can be used to predict effective intensity, and the predictive brain patterns for these types of effects are uncorrelated with the NPS[22–24]. Further, these studies showed limited evidence for the generalizability of the NPS to other types of somatic pain (mechanical, electrical, and laser)[24,25]. However, whether the NPS generalizes to capturing visceral pain has not been systematically tested. This is crucial to determine (a) whether the NPS captures common core responses across diverse types of evoked pain, particularly visceral and somatic pain, and (b) whether there are systematic differences between visceral and somatic pain that cannot be captured by the NPS. Jointly, these goals can inform us of whether the pain is a multi-component or multidimensional process at the neural level, and what the essential neural "ingredients" might be for different types of pain.

In this work, we show that the NPS responds robustly to both somatic and visceral aversive stimulation, and correlates with the subjective visceral pain experience. This identifies the NPS as a "common core pain system" that generalizes across pain types, including visceral stimulation. We also show that, contrary to the NPS, existing signatures for nonpainful affective processes (negative emotion, social rejection, and vicarious pain) do not respond consistently to somatic nor visceral stimulation. This demonstrates the sensitivity of the NPS to pain versus other affective processes and implies that visceral pain does not activate more "emotional" brain patterns compared to somatic pain, as commonly assumed. Next, we show that the NPS reliably discriminates individual brain responses to painful stimuli from nonpainful affective and cognitive manipulations across a sample of independent studies. This establishes the specificity of the NPS to pain. Finally, we train and validate a network-based classifier that reliably discriminates thermal cutaneous stimulation from rectal distension as prototypical examples of somatic and visceral pain, respectively, both in cross-validation and independent cohorts. Esophageal pain shows a pattern similar to somatic pain, while gastric and vulvar pain shows an intermediate pattern. This finding highlights differences in the neural basis of different types of visceral and somatic pain beyond the NPS as a common core pain system, thereby providing a basis for future brain-based models of visceral pain perception[26].

## Results

**Expression of the neurologic pain signature during visceral and somatic stimulation (NPS generalizability).** We tested whether the NPS responded similarly to both aversive/painful visceral and somatic stimulation. To this end, we computed the expression of the NPS for contrasts of aversive stimulation versus nonpainful stimulation or aversive stimulation versus baseline (depending on the study; see "Methods" for details). NPS expression in the four visceral pain/discomfort studies is shown in Fig. 1a, with the highest response in the esophageal pain study (Study 5). NPS response was $0.043 \pm 0.009$ $[t(14) = 4.66, P = 0.0004, d_a = 1.20 \, [0.423, 1.977]$, accuracy $= 86.67\%]$ for Study 1 (gastric pain); $0.045 \pm 0.010$ $[t(14) = 4.69, P = 0.0003, d_a = 1.21$ $[0.432 \, 1.988]$, accuracy $= 93.33\%]$ for Study 2 (rectal discomfort); $0.053 \pm 0.007$ $[t(28) = 8.08, P < 0.0001, d_a = 1.50 \, [0.917, 2.083]$, accuracy $= 96.6\%]$ for Study 3 (rectal discomfort), and $0.122 \pm 0.006$ $t(29) = 20.49, P < 0.0001, d_a = 3.74 \, [2.901, 4.579]$, accuracy $= 100\%]$ for Study 5 (esophageal pain). These effect sizes ($d_a$) are 1.5–5 times larger than typical "large effects"[27]. The accuracy statistics reported above indicate that the NPS can

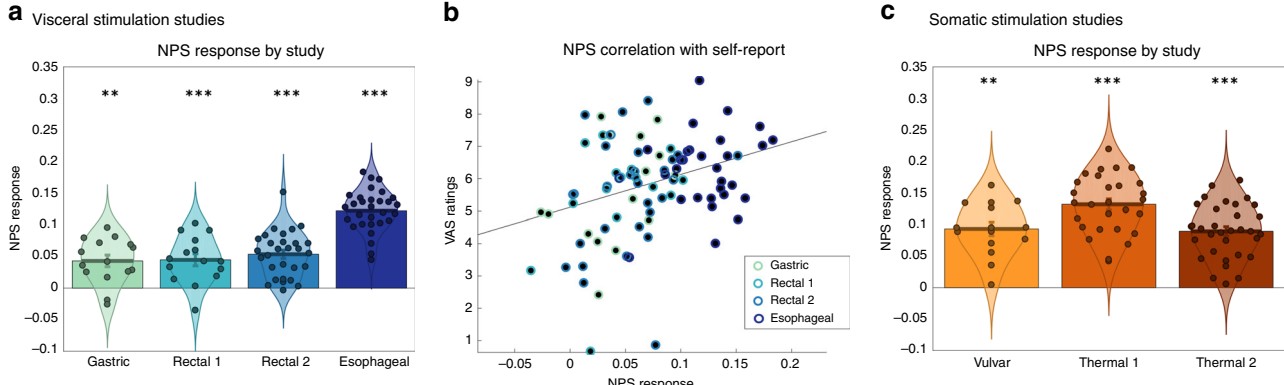

**Fig. 1 Response of the neurologic pain signature (NPS) to different types of aversive/painful stimulation. a** NPS response to visceral stimulation types. Units reflect the cosine similarity of activation images and the NPS (see "Methods" for details). Each circle indicates the NPS response for an individual subject (Ns = 15, 15, 29, and 30). Bar plots (mean ± standard error) and violin plots of NPS responses for each visceral study. **b** Scatterplot of the relationship between the average NPS response and average visual analog scale (VAS) ratings of experienced pain (after accounting for the effect of study). **c** NPS response to somatic stimulation types (Ns = 15, 28, and 33) units reflect cosine similarity. Bar plots (mean ± standard error) and violin plots of NPS responses for each somatic study. **P < 0.01, ***$q_{FDR}$ < 0.05 (two-tailed, one-sample t test). Source data are provided as a Source Data file.

accurately detect which of the two conditions is noxious for an individual participant in a forced-choice test.

Robust regression analysis demonstrated that the NPS response significantly predicted pain/discomfort intensity VAS ratings, even though the stimulus intensity was individually calibrated ($r_{weighted}$ = 0.40, $\beta_{robust}$ = 12.44 ± 04.45, P = 0.0064; Fig. 1a). Individual samples for visceral studies were correlated r = 0.38, r = 0.20, and r = 0.30 for gastric and the two rectal studies, respectively. We note that sample sizes varied from N = 15–29, and therefore individual-study correlations are likely to be highly unstable[28]. A full treatment of individual differences in pain responses remains for future larger-sample studies.

The NPS response in the 3 somatic pain studies is shown in Fig. 1b. NPS response was 0.093 ± 0.011 [t(14) = 8.61, P < 0.0001, $d_a$ = 2.22 [1.310, 3.130], accuracy = 100%] for Study 4 (vulvar pain); 0.132 ± 0.009 [t(27) = 15.22, P < 0.0001, $d_a$ = 2.88 [2.132, 3.628], accuracy = 100%] for Study 6 (thermal pain) and 0.089 ± 0.007 [t(32) = 12.11, P < 0.0001, $d_a$ = 2.11 [1.508, 2.712], accuracy = 100%] for Study 7 (thermal pain).

Overall, these results indicate that the NPS responded robustly to both somatic and visceral pain, and correlated with the subjective visceral pain experience. Comparing effect sizes, effects are larger in thermal, vulvar, and esophageal pain than rectal and gastric pain, suggesting that while the NPS generalizes, systems beyond the NPS may also be important for gastric and rectal pain.

**Expression of other affective signatures during visceral and somatic stimulation (NPS sensitivity).** Because visceral pain is often thought to evoke stronger affect than somatic pain, we additionally tested whether brain responses to visceral and somatic stimulation were more similar to the NPS than other brain-based markers that track self-reported differences in social rejection[23], picture-induced negative emotion[22], and vicarious pain[24].

As illustrated in Fig. 2a, overall studies together, data acquired during pain correlated significantly and positively with the NPS (average within-person Pearson spatial correlation r = 0.074 ± .0030 (SE), t(164) = 24.41, $d_a$ = 1.92 [1.662, 2.177], P < 0.0001, $q_{FDR}$ < 0.05). This association was significant for each individual study, across both somatic and visceral types (Fig. 2b). Turning to signatures for other affective processes in healthy individuals, data acquired during pain also correlated significantly and positively with the neural signatures for social rejection

(r = 0.017 ± 0.0019, t(164) = 8.77, $d_a$ = 0.697 [0.526, 0.866], P < 0.0001, $q_{FDR}$ < 0.05) and negative emotion (PINES) (r = 0.006 ± 0.0019, t(164) = 3.45, $d_a$ = 0.246 [0.091, 0.4004], P = 0.0007, $q_{FDR}$ < 0.05), demonstrating some match to these other patterns, but with associations 4 and 12 times weaker, respectively, than associations with the NPS (for full details, see Supplementary Table S1). In addition, the positive associations with these patterns were not consistent across pain types; associations with the rejection pattern were driven by one rectal and the thermal studies, and associations with the PINES were driven by esophageal and vulvar studies (Fig. 2f). A significant negative correlation was found with the signature for vicarious pain (r = −0.012 ± 0.0021, t(164) = −5.78, $d_a$ = −0.445 [−0.604, −0.284], P < 0.0001, $q_{FDR}$ < 0.05), indicating lack of a positive match to this pattern. The correlation with the NPS was significantly stronger than the correlations with the three other neural signatures (ANOVA F(3,492) = 286.23, P < 0.0001; P < 0.0001 for all three pairwise comparisons versus NPS after Bonferroni correction for multiple testing).

**Expression of the neurologic pain signature during pain, negative emotion, and cognitive control (NPS specificity).** Because the studies used to examine commonalities between somatic and visceral pain in the analyses reported above differ in terms of scanning hardware, pulse sequences, participant demographics, and experimental designs, we conducted a separate validation to evaluate whether the NPS can reliably discriminate individual brain responses to painful stimuli from nonpainful control manipulations across a sample of independent studies (i.e., specificity of the NPS), with 6 studies manipulating pain (by thermal, mechanical, or rectal stimulation) and 12 control studies involving either manipulations of cognitive control or negative emotion (see ref. [31] for details). This analysis revealed that the NPS could effectively discriminate brain responses to different kinds of pain from conceptually related experimental manipulations (AUROC = 0.93, sensitivity = 73%, specificity = 92%, accuracy = 86 ± 2.1%, $d_a$ = 2.13 [1.804, 2.449], Fig. 3).

**Voxel-wise analysis of responses common to aversive somatic and visceral stimulation.** To identify brain regions that exhibited similar changes in fMRI activation during somatic and visceral stimulation, we performed a voxel-based conjunction analysis of visceral and somatic stimulation modalities (each thresholded at

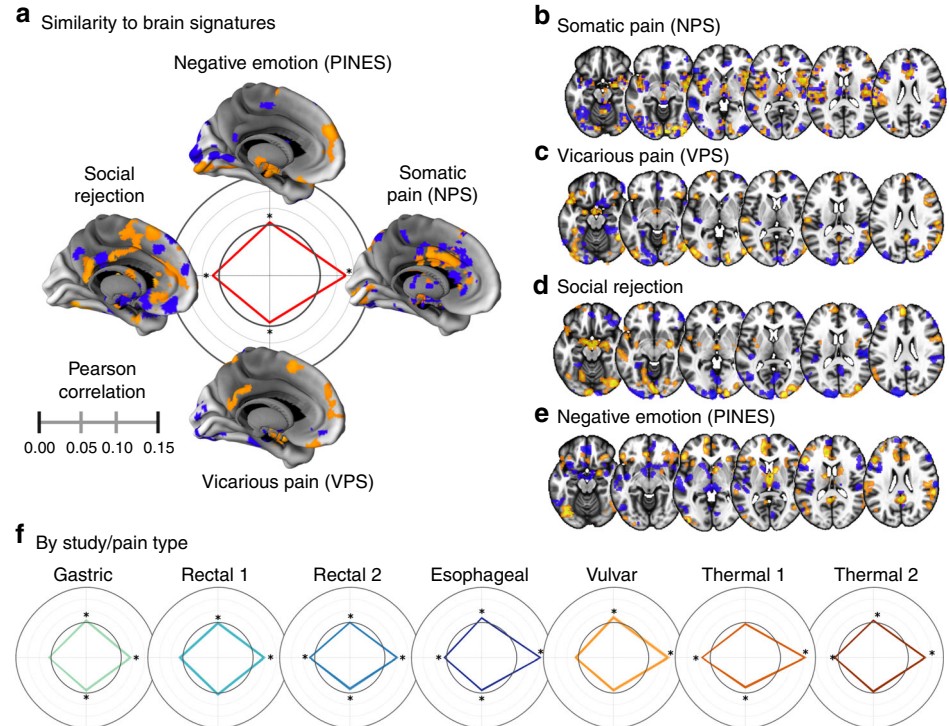

**Fig. 2 Spatial similarity between responses to different pain types and neurologic signatures. a** The similarity between brain responses pooled across pain types. Pearson correlations between pooled brain responses and neural signatures for somatic pain (NPS), vicarious pain (VPS), social rejection, and negative emotion (PINES). Lines show the mean association across participants, with shaded standard errors. The inner gray circle reflects a correlation of zero; points inside the circle are negative associations, and outside the circle are positive associations. Brain maps depict the 10,000 most positive and 10,000 most negative model weights on the cortical surface of the left hemisphere[29] and the basal ganglia[30]. These weights are shown for display purposes only to indicate the brain regions that have the greatest influence in making predictions. Positive weights are shown in warm colors and negative weights in cool colors. *$P < 0.05$ (two-tailed, one-sample $t$ test). **b**–**e** A visual depiction of the four neural signatures, with the same threshold as in (**a**). Warm colors indicate where increased activity leads to more positive outcomes and cool colors indicated where decreased activity leads to more positive outcomes. **f** Correlations between the average activity for each study. Scale and axes are identical to those in panel (**a**). *$P < 0.05$ (two-tailed, one-sample $t$ test). Source data are provided as a Source Data file.

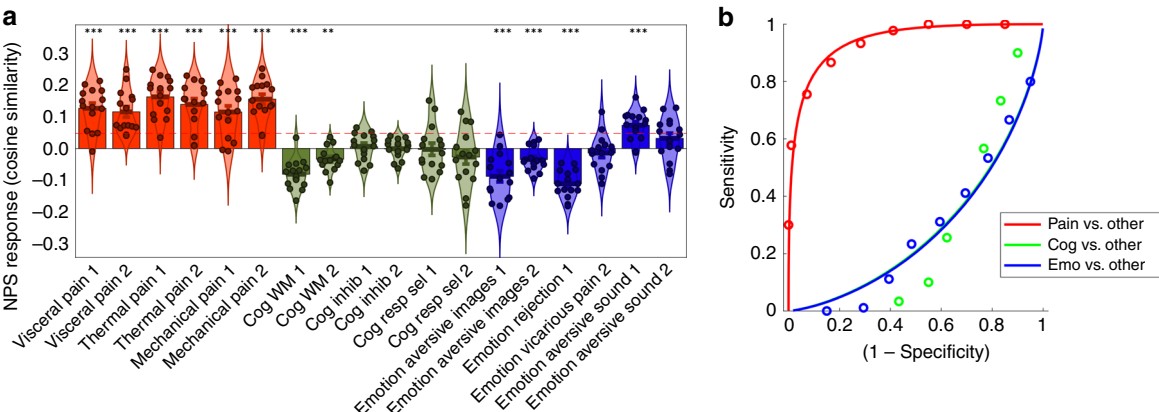

**Fig. 3 Neurologic pain signature (NPS) expression discriminates brain responses to painful and nonpainful experimental manipulations. a** Bar plots (mean ± standard error) and violin plots of NPS expression for 18 studies ($N = 15$ for each study), including experiments that evoked painful sensations (red) or manipulated cognitive control (green) or negative emotion (blue). Each circle indicates NPS expression (quantified using cosine similarity) for an individual subject. The dashed line indicates the cutoff that maximizes overall accuracy in pain/no-pain classification. *$P < 0.05$, **$P < 0.01$, ***$q_{FDR} < 0.05$ (two-tailed, one-sample $t$ test). **b** Receiver-operator characteristic curves of NPS classification of pain (red), cognitive control (green), and negative emotion (blue). Circles indicate deciles of NPS response and solid lines show the Gaussian model fits. Source data are provided as a Source Data file.

$q < 0.05$ false discovery rate (FDR)-corrected[32]. For each of the visceral ($N = 89$) and somatic subsets ($N = 76$), we controlled for the effect of the study using covariates for differences across studies (four visceral studies and three somatic studies, total $N = 165$, see "Methods" for details).

The conjunction analysis revealed that regions activated by somatic (Fig. 4a and Supplementary Table S2) and visceral stimulation (Fig. 4b and Supplementary Table S4) overlapped in a distributed set of brain regions (Fig. 4c and Supplementary Table S6), including midbrain, cerebellum, lentiform

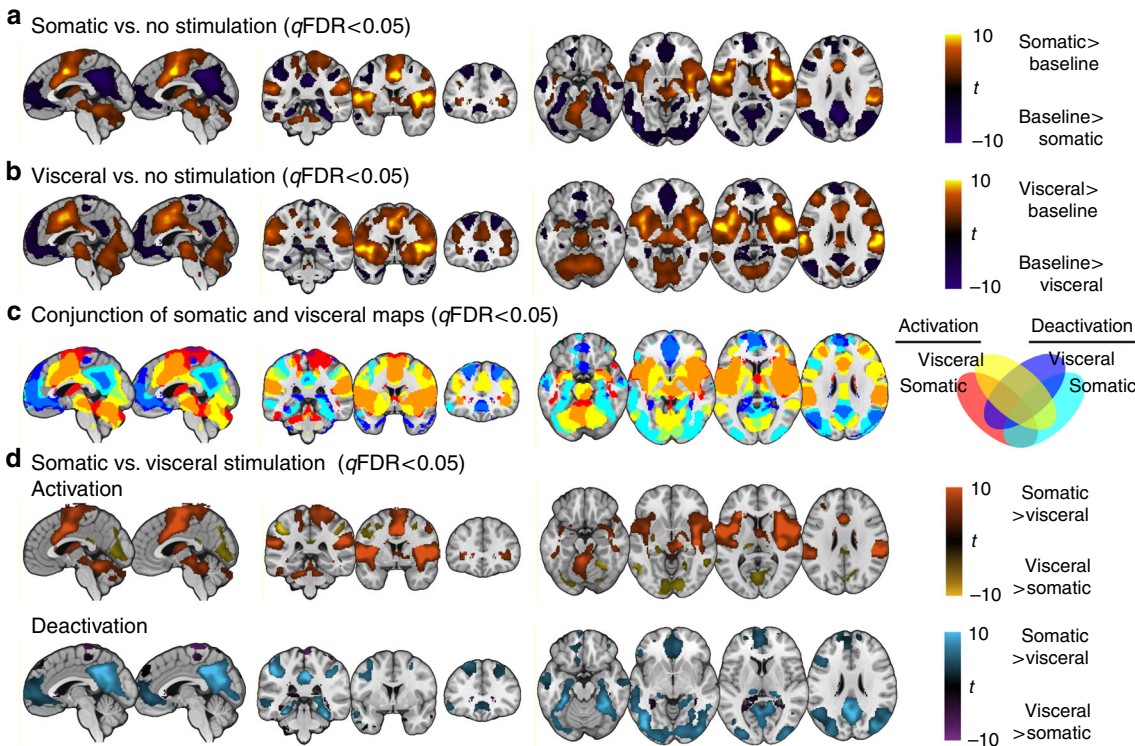

**Fig. 4 Common and distinct fMRI activation for somatic and visceral pain, assessed using univariate voxel-based GLMs.** Parametric group-level maps of responses across participants and studies, controlling for interstudy differences. **a** The contrast of somatic stimulation versus baseline. Warm colors correspond to increased levels of activation and cool color deactivation (N = 76). **b** The contrast of visceral stimulation versus baseline (N = 89). **c** The conjunction of activation and deactivation maps for somatic and visceral stimulation (N = 165; the intersection of significant somatic and visceral results). **d** Contrasts of somatic (N = 76) and visceral pain (N = 89). In the top panel, effects are masked with the map of somatic pain > baseline for orange areas and visceral pain > baseline for yellow areas. Thus, the resulting maps show stimulus-related increases that are stronger for somatic (orange) or visceral (yellow). In these bottom panels, the difference map is masked with somatic pain < baseline for cyan areas and visceral pain < baseline for purple areas. Thus, the resulting maps show areas where stimulus-related deactivation occurs and is stronger in somatic pain (cyan), and deactivation occurs and is stronger in visceral pain (purple).

nucleus (putamen and pallidum), hypothalamus, thalamus (ventral lateral and ventral posterior lateral nucleus), para-hippocampal gyrus/entorhinal cortex, insula (posterior, middle, and anterior parts), postcentral gyrus (including a medial cluster in primary somatosensory cortex (SI), and a ventral lateral cluster, including parietal/Rolandic operculum/ secondary somatosensory cortex (SII)), adjacent inferior parietal lobule, superior temporal gyrus, and inferior frontal gyrus (ventrolateral prefrontal cortex (vlPFC)), lateral pre-central gyrus, including primary motor cortex (MI) and premotor cortex, anterior and posterior midcingulate cortex (aMCC, pMCC) and adjacent medial frontal gyrus, and superior/middle frontal gyrus (dorsolateral prefrontal cortex (dlPFC)).

The conjunction analysis also revealed that regions *deactivated* by somatic (Fig. 4a and Supplementary Table S3) and visceral stimulation (Fig. 4b and Supplementary Table S5) overlapped in multiple brain regions (Fig. 4d and Supplementary Table S7), including thalamus (pulvinar), hippocampus, parahippocampal gyrus/perirhinal cortex, temporal pole, rostral middle/inferior temporal gyrus, occipital cortex and adjacent caudal middle and superior temporal gyrus, perigenual and subgenual anterior cingulate cortex (pACC, sACC) and adjacent medial, middle and superior frontal gyrus (vmPFC and dorsomedial prefrontal cortex (dmPFC)), lateral middle and superior frontal gyrus (dlPFC), PCC and adjacent precuneus, superior parietal lobule, left dorsal precentral gyrus (MI, premotor cortex), and left dorsal postcentral gyrus (SI).

**Development and validation of a network-based classifier that differentiates visceral and somatic stimulation**. To further test if representations of the different types of somatic and visceral aversive stimulations differ at a broader spatial scale, we next assessed whether brain responses to different types of visceral and somatic stimulation were differentially correlated with seven canonical resting-state cortical networks[33]. This approach pro-vides inferences about whether each of these networks is activated on average during each pain type. Detailed results of this analysis (based on point-biserial correlations) are provided as Supple-mentary Information (Supplementary Results, Supplementary Table S8). Broadly, they revealed some commonalities across all pain types, including activation of the "ventral attention" network and deactivation of the "default network" (Fig. 5a). (The network names are based on Yeo et al.[33], and by using them, we do not imply that their function is limited to or primarily related to the label). They also revealed differences: the "somatomotor" network was activated in some pain types (thermal, vulvar, and esophageal) but deactivated in others (rectal). The "frontoparietal" network was similarly activated by some (rectal, esophageal), but deacti-vated in others (thermal). Beyond these variations in which net-works were activated and deactivated, there were also variations in the degree to which each network was activated or deactivated.

The variations noted above could serve as the basis for a brain-based classifier for somatic versus visceral pain capable of making accurate predictions about new individual participants. Many pain types (e.g., esophageal, gastric, vulvar) may potentially include both somatic and visceral elements and cannot be defined

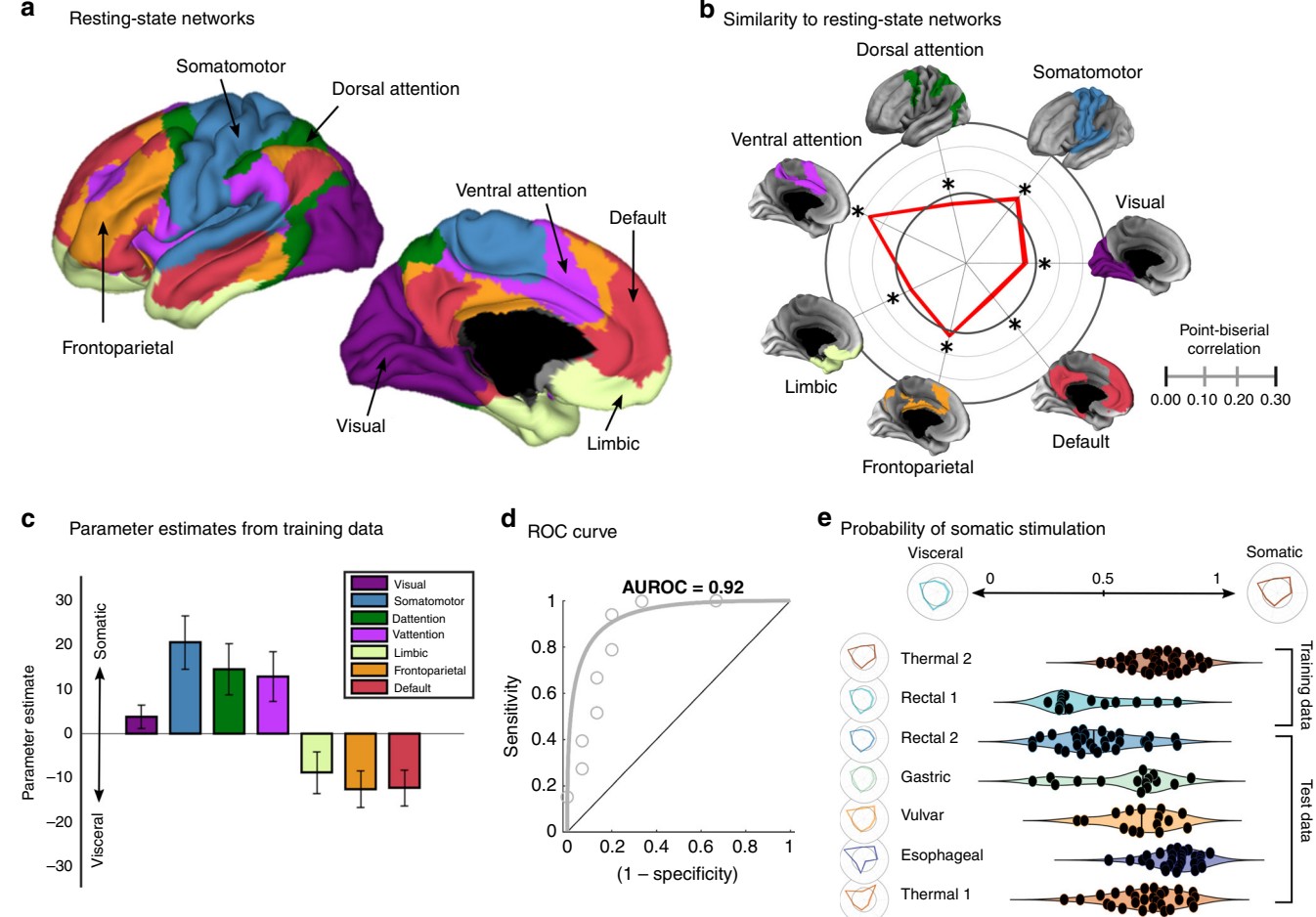

**Fig. 5 Classification of brain responses based on point-biserial correlations with resting-state networks. a** A visual depiction of the resting-state networks based on data from 1000 subjects[33]. **b** Point-biserial correlations between pooled brain responses (both somatic and visceral studies) and the seven resting-state networks. The inner bold line is the zero point, and values inside the inner circle reflect negative correlations. **c** Parameter estimates from a logistic regression model predicting whether brain responses were observed during visceral (rectal; $N = 15$) or somatic (cutaneous; $N = 33$) stimulation. Error bars reflect the standard error of the mean. *$P < 0.05$, two-tailed, one-sample $t$ tests. **d** The Receiver Operating Characteristic (ROC) curve shows cross-validated performance on the training datasets (AU = area under the curve = 0.92). The solid line shows a Gaussian model fit. **e** Generalization tests show the probability of data from novel subjects as being classified as somatic or visceral. Scores for the training datasets were estimated using cross-validation. Test data scores apply the model prospectively, with no parameter adjustment. Each circle indicates the prediction for a single subject. Polar plots depict spatial similarity with resting-state networks as in panel (**b**). Each polar plot depicts the average similarity for each study: the rectal and thermal stimulation studies used for training the classifier, and the five hold-out studies used to test its generalizability. Source data are provided as a Source Data file.

as purely somatic or visceral a priori. Therefore, we first trained a logistic regression-based pattern classifier to discriminate between the clearest examples of somatic and visceral aversive stimulation available: cutaneous thermal stimulation (Study 7) and rectal distension with an inflatable balloon (Study 2). The data used for the classification were the spatial correlations of each individual participant's stimulus-induced activity map with each of the seven resting-state networks. We tested classification accuracy on out-of-sample participants in Studies 2 and 7 using ten-fold cross-validation. Then, we tested the classifier's performance on data for all pain types in independent studies, including thermal (Study 6) and rectal (Study 3) studies, and gastric (Study 1), vulvar (Study 4), and esophageal (Study 5) studies.

The classifier exhibited high levels of discriminability for thermal versus rectal when applied to out-of-sample participants in cross-validation (AUROC = 0.92, sensitivity = 94%, specificity = 80%, balanced accuracy = $87 \pm 4.4\%$, $d_a = 2.22$ [1.46, 2.98]). As illustrated in Fig. 5b, c and Supplementary Table S8, somatic stimulation was indicated by a combination of (a) more positive

correlations with the somatomotor network (logistic regression ($\hat{\beta} = 19.89 \pm 5.94$, $P = 0.0008$), driven by a positive correlation in thermal versus negative in rectal; (b) positive correlations with the dorsal attention network ($\hat{\beta} = 14.21 \pm 5.72$, $P = 0.013$) driven by zero correlation in thermal versus a negative correlation in rectal; and (c) more positive correlations with the ventral attention network ($\hat{\beta} = 12.54 \pm 5.50$, $P = 0.023$), driven by a more strongly positive correlation in thermal. On the other hand, visceral stimulation was indicated by (a) more positive correlations with the frontoparietal network ($\hat{\beta} = -12.35 \pm 4.09$, $P = 0.0025$), driven by a significant positive correlation in rectal versus a near-zero negative correlation in thermal; and (b) more positive correlations with the default network ($\hat{\beta} = -11.89 \pm 4.02$, $P = 0.0031$), driven by a near-zero negative correlation in rectal versus a strong negative correlation in thermal. More positive correlations with the limbic network exhibited a nonsignificant trend towards predicting visceral stimulation ($\hat{\beta} = -8.66$, $P = 0.063$, driven by a zero correlation in rectal versus a nonsignificant negative correlation in thermal).

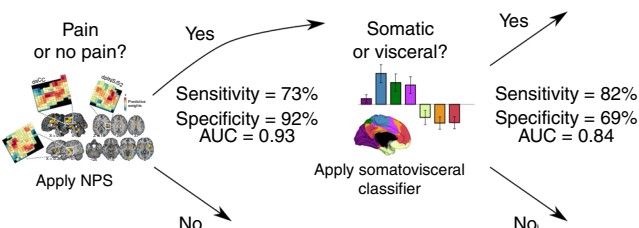

Two-stage pain classification

**Fig. 6 Overview of the proposed two-stage pain classification process based on the potential biomarkers validated and developed in this paper.** First, the NPS is applied as a classifier for pain/no pain. Subsequently, if the decision is "pain", a somatovisceral pain classifier is applied to determine the pain modality/type. AUC, area under the receiver operating characteristic curve in validation tests. AUC is a threshold-free measure of accuracy. Visualization of the NPS adapted from Woo et al.[20] with permission.

Testing the pattern classification model on the remaining studies, which include gastric (Study 1), rectal (Study 3), vulvar (Study 4), esophageal (Study 5), and thermal (Study 6) stimulation, revealed a continuum of somatovisceral expression. The classifier demonstrated successful prospective generalization to new studies when tested on the rectal (Study 3) and thermal (Study 6) test sets (AUROC = 0.84, sensitivity = 82%, specificity = 69%, balanced accuracy = $76 \pm 5.7\%$, $d_a = 1.34$ [0.765, 1.915]). The esophageal ($P_{somatic} = 96.67 \pm 3.28\%$, $P < 0.0001$) and vulvar ($P_{somatic} = 86.67 \pm 8.78\%$, $P = 0.083$) studies were largely classified as somatic (although the latter only exhibited a nonsignificant trend), whereas the gastric ($P_{somatic} = 66.67 \pm 12.17\%$, $P = 0.97$) study exhibited an intermediate classification rate (Fig. 5e).

**Voxel-wise analysis of differential responses to aversive somatic and visceral stimulation**. To identify brain regions that exhibited differential levels of fMRI activity for visceral and somatic stimulation, we conducted a standard voxel-wise general linear model (GLM) analysis comparing stimulation modalities while controlling for the effect of individual studies (four visceral studies and three somatic studies, total $N = 165$, see "Methods" for details).

Of the regions that showed significant activation (i.e., increased activity to somatic or visceral stimulation compared to baseline), regions demonstrating significantly stronger activation during somatic compared to visceral stimulation included midbrain, cerebellum, basal ganglia (caudate, lentiform nucleus, claustrum), thalamus (midline/medial dorsal/anterior nuclei), insula (posterior, middle, and anterior), inferior parietal lobule, postcentral gyrus/SI, parietal operculum/SII, temporoparietal junction, aMCC/pMCC, (pre)motor cortex, and middle frontal gyrus (dlPFC) (Supplementary Table S9 and Fig. 4a, b, e). Regions more strongly activated by visceral stimulation included thalamus (pulvinar), lentiform nucleus, amygdala, (caudal) parahippocampal gyrus, fusiform gyrus, occipital cortex (lingual gyrus, cuneus), precuneus/PCC, middle temporal gyrus, inferior and superior parietal lobule, and precentral/middle frontal gyrus (details in Supplementary Table S10 and Fig. 4a, b, e).

Of the regions that showed significant deactivation (i.e., decreased activity to somatic or visceral stimulation compared to baseline), regions demonstrating significantly stronger deactivation during somatic compared to visceral stimulation included rostral hippocampus/parahippocampal gyrus/entorhinal cortex, fusiform gyrus, temporal pole, middle and inferior temporal gyrus, occipital cortex (inferior & middle occipital gyrus, lingual

gyrus, cuneus), pACC, sACC, vmPFC, orbitofrontal cortex (OFC) and dmPFC, PCC, precuneus, superior and inferior parietal lobule, vlPFC, and right dlPFC (Supplementary Table S11 and Fig. 4a, b, f). A more limited number of regions showed significantly stronger deactivation during visceral compared to somatic stimulation, including caudal hippocampus/parahippocampal gyrus and adjacent fusiform gyrus, caudate, right precentral/medial frontal gyrus, and right superior parietal lobule (Supplementary Table S12 and Fig. 4a, b, f).

## Discussion

Understanding the common and distinct brain representations underlying visceral and somatic pain is critical for assessing the neurophysiological mechanisms underlying different forms of pain. While previous studies have pointed to both commonalities and differences[10], this study identifies brain-wide commonalities that generalize across studies and types of painful stimulation, and brain network-level changes that are robust enough to permit brain-based classification of visceral versus somatic pain in independent participants. These findings can be integrated in a two-stage classifier that can be applied to future studies (Fig. 6). In the first step, the NPS is applied to accurately discriminate pain from nonpainful cognitive and affective processes. If a brain representation is identified as pain-related by the NPS, our somatovisceral pain classifier can be applied to accurately discriminate visceral from somatic pain. This demonstration and model involved several key findings, which we review below.

First, the NPS responded to all types of visceral stimulation and predicted individual visceral pain/discomfort ratings, confirming the NPS as a common "core pain-related network" that generalizes across modalities. Voxel-wise GLM conjunction analyses corroborate this: the vast majority of key NPS regions were shown to be commonly responsive to somatic and visceral stimulation. Further, the NPS captured visceral and somatic pain/discomfort better than several published neural signatures for non-nociceptive affective processes (rejection, vicarious pain, and negative emotion), indicating that the neural basis of visceral discomfort/pain has more in common with somatic pain than with other affective processes. The NPS also accurately discriminated brain responses to pain from nonpainful affective and cognitive experimental manipulations. Together, these results support the sensitivity, generalizability, and specificity of the NPS to pain.

However, we also observed variation in brain representation across pain types. First, not all types of pain activated the NPS equally strongly; for example, thermal and esophageal pain produced stronger NPS responses than gastric and rectal visceral stimulation. While it was impossible here to equate the stimulus timing and subjective pain intensity across these pain types (see below), these findings suggest variation in brain responses across pain types. Findings from network-level analyses also identified differences in the pattern of networks engaged across visceral and somatic pain. Increased engagement of the somatomotor network for somatic (i.e., cutaneous thermal) pain versus the frontoparietal network for visceral (i.e., rectal) stimulation provided a double dissociation between these stimulation types. Such findings generally preclude arguments that brain differences result from one pain type being less intense or less reliable than others, and instead point to genuine processing differences. These findings were robust enough to allow brain-based classification of whether, in a particular test condition, an individual experienced somatic or visceral pain with 94% sensitivity and 80% specificity (and 82% sensitivity, 69% specificity when applied to novel studies/cohorts).

This network-based classification model permitted an examination of pain types that, although heuristically classified as visceral or somatic, may contain elements of both. Esophageal conditions stimulated the distal (i.e., smooth muscle, viscerally innervated) part of the esophagus. However, it was classified as somatic (i.e., similar to cutaneous thermal pain). By contrast, painful vulvar stimulation is sometimes considered somatic, but both gastric and vulvar stimulation fell part-way between somatic and visceral (i.e., rectal) outcomes in their brain patterns. These types of pain may involve a mixture of different pain-related brain representations, with variation in the mixture across individuals.

Though they have not tested multivariate measures like the NPS or brain-based classifiers, previous univariate findings show both convergence with and divergence from our findings. Strigo et al. ($N = 7$) found similar regions activated by painful visceral (esophageal) and cutaneous (heat on the chest) stimuli. However, they observed a stronger anterior insula response to cutaneous pain, in line with our findings, and stronger motor cortex and MCC response to visceral pain, which we did not observe. Dunckley et al. ($N = 10$) found stronger deactivations in response to visceral (rectal) compared to cutaneous (heat on left foot/ midline lower back) pain matched for perceived unpleasantness in the pACC, vmPFC, and PCC, which we did not find. Further, they found stronger anterior insula activation per intensity rating unit in visceral compared to cutaneous pain[13], which was not found by Strigo et al. or the present study. We did not calculate activation per intensity rating unit, which may account for this discrepancy. In another study ($N = 10$), Dunckley et al. found increased activation in the midbrain (nucleus cuneiformis) in visceral (rectal electrical stimulation) versus somatic (lower abdominal electrical stimulation) pain, matched for perceived intensity, which again differs from present findings. We observed a higher midbrain response to somatic pain. More recently, Koenen et al. compared brain responses to intensity-matched painful rectal distension and cutaneous thermal stimulation in 22 healthy women[10]. During the ascending phase of the stimulus, visceral pain induced greater activation in SI, dorsal and ventral anterior insula, a/pMCC, and midbrain when compared with somatic pain. This differs from our findings, as we found stronger responses to somatic pain in all these regions. Koenen et al. also found that somatic pain induced stronger dlPFC and posterior insula activation, in line with our results, but also stronger hippocampal activation, which we did not. Further, they found common activation responses in inferior parietal lobule (as we found), but also in dlPFC and vlPFC (which were differentially (de)activated by both modalities in our study), and vmPFC (which was deactivated for both somatic and visceral pain in this study).

This variability reinforces known issues with small studies and suggests that larger-sample sizes—which are approximately an order of magnitude larger in our study ($N = 165$) than most previous studies ($N = 7–22$)—are needed to identify reproducible patterns. Differences across studies could be related to stimulation procedure (fixed intensity versus individually titrated, level of intensity, stimulation type), fMRI task design, and differences in sex distribution, among other factors. However, our study suggests that it is possible to identify a common generalizable brain basis for somatic and visceral pain/discomfort, as well as reproducible pain type-specific brain processes, in spite of interstudy variation. Therefore, we believe that the differences in stimulation type, duration, etc., across the studies included here, represent a largely inevitable limitation in some respects (see below), but also constitute a strength from a generalizability perspective, as we demonstrate common activation patterns across variations in these parameters. This is especially important because

generalizability is seldom assessed and is one of the core aims of this paper.

In addition, our results partially confirm the historical assumption that somatic and visceral aversive stimulation are represented in "lateral" and "medial" pain systems, respectively, but also offer a more nuanced view. We found stronger responses to somatic stimulation in some parts of the "lateral" system (SI/ SII and posterior insula, somatomotor network), but we also found stronger responses to somatic stimulation in parts of the medial system (aMCC and anterior insula). Visceral stimulation-induced greater activity in other medial regions not typically associated with nociception, including vmPFC and other "default mode" and "limbic" regions. "Default mode" regions have recently been identified as contributing to aspects of evoked and clinical[23,34,35] pain, and the modulation of pain in animal models[36,37] via frontostriatal pathways[36]. However, visceral stimulation also induced greater activity in lateral "frontoparietal network" regions as well. These findings suggest that both an expansion and a refinement of the cortical circuits that contribute to different types of pain, and pain in different individuals, is needed—and that the field may be ready to move beyond the traditional medial/lateral pain system distinction.

Our study also has implications for the function of the somatomotor network and its role in pain. More positive activation of the somatomotor network was the strongest predictor of cutaneous vs. rectal stimulation, and it likely contributed importantly to the classification of esophageal pain as somatic. These results are consistent with previous meta-analyses showing sensorimotor activation by cutaneous (mostly thermal) pain[6,38,39]. A meta-analysis of rectal distension studies, on the contrary, did not show somatosensory activation[40], and evidence on gastric distension is mixed[41]. Esophageal stimulation studies, on the contrary, have consistently reported responses in the sensorimotor cortex[42,43]. These differences may not be inherent to the stimulus type, but rather the responses engaged. Somatosensory and "ventral attention" areas may be preferentially involved in reorienting attention towards salient external somatic stimuli[44]. Esophageal pain is brief, with relatively rapid onset, and thus likely loads highly on these processes.

On the other hand, greater activation in the "frontoparietal" network was the strongest network-level predictor in the classifier distinguishing rectal from cutaneous stimulation. This network includes anterior PFC, dlPFC, dmPFC, anterior insula, and inferior parietal lobule[45]. These regions consistently respond to aversive rectal distension[40], but also to somatic pain[6,38]. This network, which is anatomically positioned to integrate information from the "dorsal attention" network (regulating externally oriented attention in a top-down fashion) and the "default mode" network (involved in internally oriented modes of cognition, including memory retrieval and assessment of self-relevance) has been involved in executive functioning, including cognitive control, decision-making, and response selection[45]. It is also engaged during the discrimination of ambiguous, non-familiar stimuli[45]. Visceral pain may activate this network more strongly because it is more interoceptive, more goal-relevant, or more novel; future research is needed to understand its mapping with cognitive and affective processes. Greater interoceptive load in visceral pain may also account for the reduced "default-mode network" deactivation we observed for rectal versus cutaneous stimulation.

This paper has several limitations that should be addressed. The stimulation procedures and fMRI task designs of the different visceral and somatic stimulation studies were similar, but not identical. More specifically, the duration of stimuli was different between modalities and types of stimulation, ranging from 1 s stimulations analyzed as events in Study 4 (esophageal pain) to

**Table 1 Summary of studies. All studies modeled activity during painful/uncomfortable stimulation periods.**

| Study | Stimulation | N (female) | Age (mean ± SD) | Field strength | Contrast | Voxel size |
|---|---|---|---|---|---|---|
| 1 | Gastric | 15 (10) | 31.9 ± 8.8 | 3 T | certain$_{pain}$–safe$_{nopain}$ | 2.5 × 2.5 × 2.5 mm³ |
| 2 | Rectal 1 | 15 (9) | 29.5 ± 10.5 | 3 T | certain$_{pain}$–safe$_{nopain}$ | 2.75 × 2.75 × 3 mm³ |
| 3 | Rectal 2 | 29 (15) | 22.5 ± 2.8 | 3 T | certain$_{pain}$–safe$_{nopain}$ | 2.5 × 2.5 × 2.5 mm³ |
| 4 | Vulvar | 15 (15) | 23.2 ± 1.6 | 3 T | certain$_{pain}$–safe$_{nopain}$ | 2.5 × 2.5 × 2.5 mm³ |
| 5 | Esophageal | 30 (14) | 30.4 ± 8.7 | 3 T | stimulation–rest | 3.75 × 3.75 × 3.3 mm³ |
| 6 | Thermal 1 | 28 (10) | 25.2 ± 7.4 | 3 T | high heat–rest | 3.4 × 3.4 × 3.4 mm³ |
| 7 | Thermal 2 | 33 (22) | 27.9 ± 9.0 | 3 T | high heat–low heat | 3.0 × 3.0 × 3.0 mm³ |

30 s stimulations analyzed as blocks in Study 1 (gastric pain). However, this is to a certain extent unavoidable as long mechanical esophageal stimulation is not feasible due to the induction of peristaltic effects, and shorter gastric distension stimuli are impossible due to the time needed to inflate large volume gastric balloons. Further, the intensity level at which stimulations were performed differs between studies, with individually titrated stimuli used in some studies and fixed intensity stimuli in others. Moreover, severe discomfort rather than pain threshold was used to calibrate stimulation in the rectal studies, as pain threshold can often not be reached during rectal distension in healthy subjects due to intolerable levels of urgency to defecate. These differences could account for some of the differences in brain responses found between different stimulation types in this study, and this should ideally be addressed in future studies where the similarity of stimuli is maximized. Further, studies with different stimulation intensities for each type of visceral stimulation should be performed to develop a visceral NPS (ideally, one per visceral stimulation type, based on our current results) and compare them directly with the somatic NPS. Including manipulations and/or ratings of non-pain affective processes in the same paradigm to allow within-subject comparison of responses of the different affective signature to these different types of stimuli as well as their relationship with subjective ratings constitutes another interesting future direction to replicate, validate, and extend our present results. Finally, we do not analyze sex differences here, which would require a larger sample of male and female participants and a different set of analyses. Future work is required to properly evaluate sex differences in brain responses to visceral and somatic pain. However, these limitations are outweighed by a number of strengths, including the large total sample size and direct comparison of visceral and somatic stimulation modalities, as well as different stimulation types within each of these two modalities, with the latter increasing generalizability.

In conclusion, neural representations of visceral and somatic pain/discomfort showed an important degree of overlap, confirming the involvement of a "core pain-related network" common to different modalities of pain/discomfort and types of stimulation within these modalities. This network is well characterized by the NPS, as it was responsive to different visceral (as well as somatic) stimulation types, and as it accurately discriminates pain from nonpainful cognitive and affective processes. However, visceral and somatic pain/discomfort also showed some distinct neural features, as our classifier accurately discriminated visceral from somatic pain. These findings can be summarized as a two-stage classification process (Fig. 6), with the NPS serving as a first-stage "pain/no-pain" classifier, and images classified as pain sorted into somatic and visceral types based on the second brain pattern. Further, brain responses to visceral discomfort/pain did not correlate appreciably with the neural signatures of non-pain affective processes, which challenges the often assumed "stronger affective nature" of visceral pain at the

neural level. Finally, important differences were found within the visceral modality, with responses to aversive esophageal distension being more similar to painful thermal cutaneous stimulation than to aversive rectal distension.

## Methods

fMRI data from seven studies were retrospectively aggregated to identify common and distinct brain representations for visceral and somatic pain. Standard preprocessing and mass-univariate general linear models were applied to fMRI data before they were combined for multi-study analyses. Details regarding participants, stimulation procedures, task design, MRI data acquisition, and first-level data analysis are described in the Supplementary Methods (for a summary see Table 1). All subjects provided written informed consent prior to being included in the studies.

As an additional validation, we applied the NPS to brain responses from 18 studies that manipulated processes that are conceptually related to pain (total $N = 270$, $N = 15$ per study, archival data from https://neurovault.org/collections/3324/, see ref. [31] for details). These studies include manipulations intended to evoke brain activity related to pain, cognitive control, and negative emotion. Hence, we use brain responses to pain manipulations to test the sensitivity of the NPS, and responses to "control" studies of cognitive control and negative emotion to evaluate the specificity of the NPS. To perform these tests, we computed the cosine similarity between the NPS and brain maps for each subject. This cosine similarity measure served as the basis for single-interval classification (pain vs. no pain) with a cutoff selected to maximize overall accuracy.

**Statistical analysis**. Second-level statistical models were conducted using CANlab neuroimaging analysis tools, which is an open-source toolbox written for MATLAB (see https://canlab.github.io/).

Brain responses to visceral and somatic pain were analyzed by performing a second (group)-level univariate GLM analysis, i.e., a multiple regression including the pain versus baseline/rest contrast for each of the four visceral and three somatic pain studies [certain$_{pain}$–safe$_{nopain}$ for Study 1 (gastric pain), Study 2 and 3 (rectal discomfort), and Study 4 (vulvar pain); pain–rest for Study 5 (esophageal pain); pain–rest for Study 6 (cutaneous thermal pain) and pain–nonpainful warmth for Study 7 (cutaneous thermal pain)]. Contrasts of interest for the regression included: somatic pain and visceral pain–implicit baseline, somatic pain–implicit baseline, visceral pain–implicit baseline, somatic–visceral stimulation, and visceral–somatic stimulation, thresholded at a voxel-level threshold of $q_{FDR} < 0.05$, controlling for study (via nuisance regressors).

To identify brain regions that exhibited overlapping activation and deactivation, a conjunction analysis using the minimum test statistic was performed[46]. In this approach, regions that showed either increased or decreased activation for both the somatic and visceral contrasts ($q_{FDR} < 0.05$) exhibited a significant effect. To identify brain regions that showed significant differences in activation, the somatic–visceral and the visceral–somatic contrasts were inclusively masked with the contrast of somatic and visceral > baseline. Differences in deactivation were identified using the same procedure, with an inclusive mask of somatic and visceral < baseline.

To quantify the NPS response in each of the studies, the signature response was estimated for each test subject by taking the dot product of vectorized activation images with the signature pattern, yielding a continuous scalar value[19]. This value was scaled using the l2 norms of activation images and signature patterns, to reduce differences in scaling across studies. Effect sizes ($d_a$) are reported as a continuous measure of the NPS' ability to separate pain from no pain (i.e., the difference between the average NPS response in the pain and baseline conditions, divided by the pooled standard deviation). T-statistics were computed using a one-sample $t$ test of these differences. Confidence intervals (with 95% coverage) for effect size estimates were computed using the method proposed by Hedges and Olkin[47]. Classification accuracy was estimated as the proportion of individuals who had a larger NPS response to stimulation compared to baseline conditions.

To test whether NPS responses in visceral pain studies 1–3 and 5 predicted pain ratings, we used robust regression analysis (to down weight the influence of

potential outliers), controlling for study. For this purpose, we calculated the average of the online VAS ratings of pain intensity during the certain$_{pain}$ and safe$_{nopain}$ conditions over all trials, and subtracted the ratings obtained during the safe$_{nopain}$ condition from the ratings obtained during the certain$_{pain}$ condition in each subject ($\Delta$certain$_{pain}$–safe$_{nopain}$) for studies 1–3. For the esophageal pain study (Study 5), we used the average of the pain ratings, and ratings during rest were assumed to be zero because online ratings were not collected during rest. The inference was made on the regression coefficients using a one-sample $t$ test, and a weighted correlation between NPS responses and VAS ratings was estimated using IRLS weights.

To compare the brain responses to the different pain modalities in the different studies with the NPS and other recently developed neural signatures for negative affect, social rejection, and vicarious pain as well as seven resting-state networks[33], we used whole-brain spatial correlations (based on Pearson correlation coefficients ranging from −1 to 1, with their significance tested by a one-sample $t$ test on Fisher r-to-z transformed values). A repeated-measures ANOVA was performed to confirm that these pain manipulations produced brain responses that were more similar to the NPS than to other brain signatures. In the case of the resting-state networks, which are binary masks, we more prefer these correlation coefficients as point-biserial correlations for clarity. FDR correction was used for inference, correcting across four comparisons when evaluating similarity to brain signatures and across seven comparisons for the resting-state networks.

Finally, we trained a between-subject classifier to differentiate visceral discomfort (Study 2, rectal stimulation) from somatic pain (Study 7, cutaneous heat) by entering Fisher transformed point-biserial correlation coefficients with the seven resting-state networks for each subject into a logistic regression analysis with somatic versus visceral stimulation as the binary dependent variable. To estimate the generalizability of the classifier, tenfold cross-validation was performed on the training data. To verify the stability of cross-validated models, we iterated the entire tenfold procedure 1000 times and examined the standard deviation of classification error and mean Pearson correlation of parameter estimates across all iterations. This model assessment revealed that the model was quite stable. Classification error had a standard deviation of only 1.06%, and model parameters exhibited a correlation of $r = 0.9998$. For out-of-sample tests, the average model coefficients from all cross-validation folds were applied to the remaining testing data (studies 1, 3, 4, 5, and 6). Binomial tests were used to compare the proportion of subjects being classified as either somatic or visceral in the test data (studies 1, 3, 4, 5, and 6), relative to the threshold determined from the training sample (62.49% somatic).

**Reporting summary**. Further information on research design is available in the Nature Research Reporting Summary linked to this article.

## Data availability

Data are available from the authors on request and at https://neurovault.org/collections/8707/. Data used to validate the NPS are available at https://neurovault.org/collections/3324/. Source data are provided with this paper.

## Code availability

Data were analyzed using CANlab neuroimaging analysis tools available at https://github.com/canlab/ and from https://github.com/canlab/2020_Visceral_and_Somatic_Pain.

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

## Acknowledgements

The authors would like to thank L. Aerts, R. Peeters, S. Coen, and A. Farmer for help with acquiring the data of Studies 1, 4, and 5. We thank the "IRMaGe" platform of the Grenoble University, that benefit from funding from "France Life Imaging", part of the French program "Investissement d'Avenir", for their help with the fMRI acquisitions in Study 2. We thank T. Muratsubaki and J. Morishita for their assistance with the data collection of Study 3. Studies 1 and 4 were funded by a Research Project (G.0722.12) from the Research Foundation-Flanders (FWO-Vlaanderen) to L.V.O. and J.T. Study 2 was funded by grants from the Direction de la Recherche Clinique of the Grenoble-Alpes University Hospital, and from the pharmaceutical companies, Cephalon and Ferring to B.B. Study 3 were supported by a collaborative grant from the Japanese Society for the Promotion of Science (JSPS) and the FWO-Vlaanderen (VS.014.13N) to L.V.O. and S.F. and JSPS-KAKENHI grant 26460898 to M.K. Study 6 was funded by NIH R01MH076136 and R01DA035484 to T.D.W. Study 7 was funded by NSF award 0631637 and NIH 1RC1DA028608, R01DA027794, and R01MH076136 to T.D.W.

## Author contributions

L.V.O., M.K., H.G.L, P.D., J.T., and T.D.W. designed experiments. A.R., B.B., C.D.-M., E.P., H.G.L., J.T., P.D., P.E., Q.A., and S.F. acquired and contributed neuroimaging data. H.G.L., M.K., P.D., L.V.O., A.R., C.D.-M., E.P., P.A.K., and T.D.W. conducted data analysis. L.V.O., M.K., P.A.K., and T.D.W. wrote the original draft of the paper. All authors provided feedback and revised the paper.

## Competing interests

The authors declare no competing interests.
