## [Peer Review File · Nature Communications]

Reviewers' Comments:

Reviewer #2:

Remarks to the Author:

This reviewer's comments have been adequately addressed. The study represents an important, novel, and convincing contribution to the understanding of the brain mechanisms of pain.

Markus Ploner

Reviewer #3:

Remarks to the Author:

The revised version of this manuscript is in my opinion excellent. All my comments have been addressed. This will make a significant contribution to the literature. My congratulations to the authors for the quality of their work.

Reviewer #4:

Remarks to the Author:

Review for NCOMMS-20-26197-T

In this revision, authors answered almost completely to my concerns. The manuscript is now relatively easy to read, even though discussion could be shortened again. Out of subjects assertions have been removed. A real effort has been made to simplify and not to generalize to clinical pain states. It is more understandable. I consider that Figure 3 is now very important because it closes the 10-years-length debates on alleged absence of specificity of the NPS as compared to other 'salient' stimulations. In this figure, there is no doubt that pain DO have distinctive features as compared to other 'saliency' requiring tasks, including both cognitive ones and negative emotions. This is a major advance in our understanding of pain processes and a major step in recent literature. The 93% specificity of NPS in describing pain as compared to other tasks closes old debates based on false negative findings and claiming that pain is nothing else than a universal detection system.

Reviewer #2

This reviewer's comments have been adequately addressed. The study represents an important, novel, and convincing contribution to the understanding of the brain mechanisms of pain.

Markus Ploner

We would like to thank the reviewer for his positive appreciation of our revision as well as the importance and novelty of our work.

Reviewer #3

The revised version of this manuscript is in my opinion excellent. All my comments have been addressed. This will make a significant contribution to the literature. My congratulations to the authors for the quality of their work.

We would like to thank the reviewer for their positive appreciation of both our revision and the significance and quality of our work.

Reviewer #4

In this revision, authors answered almost completely to my concerns. The manuscript is now relatively easy to read, even though discussion could be shortened again. Out of subjects assertions have been removed. A real effort has been made to simplify and not to generalize to clinical pain states. It is more understandable.

We are pleased to hear that the reviewer is satisfied with our implementation of his helpful suggestions during the previous revision round, which have indeed made the paper easier to read. To address his last remaining concern, we further reduced the length of the discussion where possible.

I consider that Figure 3 is now very important because it closes the 10-years-length debates on alleged absence of specificity of the NPS as compared to other 'salient' stimulations. In this figure, there is no doubt that pain DO have distinctive features as compared to other 'saliency' requiring tasks, including both cognitive ones and negative emotions. This is a major advance in our understanding of pain processes and a major step in recent literature. The 93% specificity of NPS in describing pain as compared to other tasks closes old debates based on false negative findings and claiming that pain is nothing else than a universal detection system.

We would like to thank the reviewer for the positive appreciation of our newly added analysis/figure, which we indeed believe adds to the value of the paper.